# DS6, Deformation-aware Semi-supervised Learning: Application to Small Vessel Segmentation with Noisy Training Data

**Soumick Chatterjee** [1]                                          SOUMICK.CHATTERJEE@OVGU.DE
**Kartik Prabhu** [1]                                                 KARTIK.PRABHU@ST.OVGU.DE
**Mahantesh Pattadkal** [1]                                   MAHANTESH.PATTADKAL@ST.OVGU.DE
**Gerda Bortsova** [2]                                               G.BORTSOVA@ERASMUSMC.NL
**Chompunuch Sarasaen** [1]                              CHOMPUNUCH.SARASAEN@OVGU.DE
**Florian Dubost** [2]                                                FLORIANDUBOST1@GMAIL.COM
**Hendrik Mattern** [1]                                            HENDRIK.MATTERN@OVGU.DE
**Marleen de Bruijne** [2,3]                                 MARLEEN.DEBRUIJNE@ERASMUSMC.NL
**Oliver Speck** [1]                                                   OLIVER.SPECK@OVGU.DE
**Andreas Nürnberger** [1]                                  ANDREAS.NUERNBERGER@OVGU.DE

[1] *Otto von Guericke University Magdeburg, Germany*

[2] *Erasmus MC, Rotterdam, The Netherlands*

[3] *University of Copenhagen, Denmark*

**Editors:** Under Review for MIDL 2021

## Abstract

The advancement of 7 Tesla MRI systems enabled the depiction of very small vessels in the brain. Segmentation and quantification of the small vessels in the brain is a critical step in the study of Cerebral Small Vessel Disease, which is a challenging task. This paper proposes a deep learning based on U-Net Multi-Scale Supervision architecture to automatically segment small vessels in 7 Tesla 3D Time-of-Flight (TOF) Magnetic Resonance Angiography (MRA) data trained on a small imperfect semi-automatically segmented dataset and was made equivariant to elastic deformations in a self-supervised manner using deformation-aware learning to improve the generalisation performance. The proposed method achieved a dice score of 80.44±0.83 while being compared against the semi-automatically created labels and 62.07 while comparing against manually segmented region.

**Keywords:** UNet Multi-scalce Supervision, 7 Tesla MRA, TOF-MRA, Imperfect ground-truth

## 1. Introduction

7 Tesla TOF-MRA is capable in depicting small vessels non-invasively and segmentation-quantification of these vessels is an important task. Many of the existing vessel segmentation techniques are capable of segmenting medium to large vessels but often fail to segment small vessels without extensive parameter tuning or by manual corrections, albeit making them time-consuming, laborious, and not feasible for larger datasets (Jerman et al., 2015).

## 2. Methodology

The proposed backbone network of the method is a modified version of U-Net Multi-scale Supervision (UNet MSS, Zhao et al. (2019)), which performs four times downsampling using Max-Pool in the contracting path, performs upsampling using interpolation in the expanding path, and uses Batch Normalisation and ReLU in each block of the network. In addition to the final segmentation output, output of two of the following layers (different scales) interpolated using nearest-neighbour to the size of the segmentation mask are also

considered for loss calculation - hence, multi-scale supervision. To enable the model to learn consistency under elastic deformations, the work by Bortsova et al. (2019) was extended by using the modified U-Net MSS. Let $\mathcal{X}$ be the set of input volumes while $\mathcal{Y}$ is the set of corresponding labels and $\mathcal{T}$ be the set of elastic transformations. The proposed network uses a Siamese architecture to learn from the original data and deformed data using its two identical branches. The first branch is fed with the tuple (x,y), where $x \sim \mathcal{X}$, $y \sim \mathcal{Y}$, while the second branch is fed with the elastically transformed volume and label $(t(\mathrm{x}),t(\mathrm{y}))$, where $t \sim \mathcal{T}$. These tuples are passed through the U-Net MSS to derive segmentation outputs $\hat{y}_1$ and $\hat{y}_2$ respectively. These outputs are compared with the corresponding labels to derive the supervised loss. Furthermore, the $\hat{y}_1$ is elastically transformed by t $\hat{y}_1$ to $\bar{y}$. Now the $\bar{y}$ is compared with $\hat{y}_2$ for computing the consistency loss in self-supervised manner.

The proposed approach was validated using the dataset by Mattern et al. (2018), which comprises of high-resolution (300 μm) 3D TOF-MRA of 11 subjects imaged at 7T and the labels for these MRA images were created semi-automatically using Ilastik (Berg et al., 2019). The dataset was randomly divided into training, validation and test sets in the ratio of 6:2:3 and was evaluated using a 3-fold cross-validation. Every MRA image volume was converted to 3D patches with dimensions $64^3$. For the calculation of the supervised loss and the consistency loss, the focal Tversky loss (Abraham et al., 2019) was used and was optimised during training using the Adam optimiser with a learning rate of 0.01, for 50 epochs. The code is available publicly on GitHub: https://github.com/soumickmj/DS6.

## 3. Evaluation

Table 1: Comparison against Ilastik labels

| Method | Frangi Filter | MSFDF Pipeline | Attention U-Net | U-Net | U-Net MSS | U-Net + Deformation | **U-Net MSS + Deformation** |
|---|---|---|---|---|---|---|---|
| **Dice Coeff.** | 51.81±3.09 | 48.35±6.34 | 76.73±0.22 | 76.19±0.17 | 79.35±0.35 | 79.44±0.89 | **80.44±0.83** |

Table 2: Comparison against a manually segmented region

| Method | MSFDF | Frangi | Ilastik | U-Net | U-Net MSS | U-Net + Deformation | **U-Net MSS + Deformation** |
|---|---|---|---|---|---|---|---|
| **Dice Coeff.** | 52.39 | 57.59 | 50.21 | 47.45 | 52.17 | 59.81 | **62.07** |

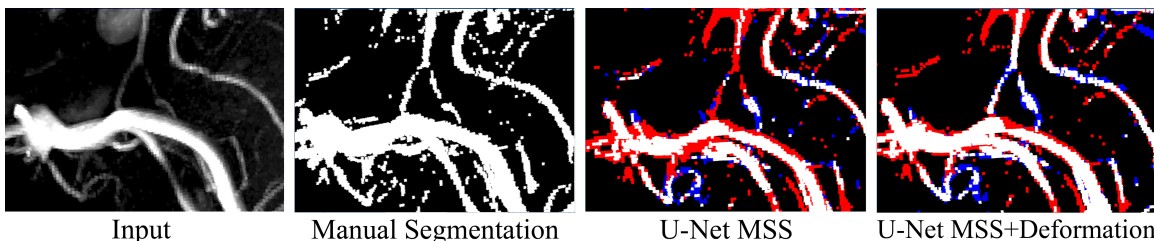

| Input | Manual Segmentation | U-Net MSS | U-Net MSS+Deformation |

Figure 1: Qualitative evaluation of various methods: Red colour indicates the false negative while blue represents false positive while being compared against manual segmentation.

Two different non-deep learning (non-DL) based methods were used for comparison - a three scale Frangi filter (Frangi et al., 1998), with a $\gamma$ of 0.1 (Frangi correction constant),

final threshold of 0.01; and the MSFDF pipeline (Bernier et al., 2018) with Otsu offset of 0.0. Comparisons were also performed against two established deep learning based baseline models - U-Net (Çiçek et al., 2016) and Attention U-Net (Oktay et al., 2018). The performance of the proposed method was compared with and without the deformation consistency. For the non-DL methods, two different pre-processing steps were performed: N4 bias field correction and brain-extraction. From the qualitative and quantitative results (Tab. 1), it can be observed that the proposed method outperformed the baseline methods. Independent two-sample t-test has shown that the incorporation of multi-scale supervision with UNet resulted in a significant improvement (p-value 0.003) and the deformation-aware learning resulted in significant improvement for both UNet and UNet MSS (p-values 0.004 and 0.013). Moreover, a single ROI was segmented manually from an input image as the labels created semi-automatically using Ilastik were imperfect and the performance of the methods was compared against this manually segmented ROI (see Tab. 2 and Fig. 1).

## 4. Conclusion

The proposed deformation-aware semi-supervised method outperformed the compared baseline methods while being trained using a small imperfectly labelled dataset and was observed that the models with deformation consistency outperformed significantly the ones without.

## Acknowledgments

This research was supported by the ESF (project no. ZS/2016/08/80646); by the Dutch Technology Foundation STW, project number P15-26, co-funded by Intel Corporation; by the ZonMw (project 104003005); and by the DFG (grant number MA 9235/1-1).

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
