# OpenReview forum: "DS6, Deformation-aware Semi-supervised Learning: Application to Small Vessel Segmentation with Noisy Training Data"
_MIDL.io/2021/Conference/Short — MIDL 2021 Poster_

### Official Review · Reviewer_tenh · 2021-04-26

**Confidence:** 3
**Final Rating:** 4

**Summary:**

The authors propose a multi-scale supervision architecture with a deformation-aware proxy task for the problem of brain vessel segmentation in MR Angiography images. To train the architecture, only a small dataset is used of semi-automatically segmented images. The proposed method outperforms several baselines, including methods with and without deep learning, and the inclusion of deformation awareness is shown to improve the performance of the method.

**Strengths:**

- The idea of using multi-scale supervision with deformation awareness for the task of small vessel segmentation is sound considering the limited availability of data for the task at hand.
- The authors compare the proposed approach both to methods that use deep learning and methods that do not. The evaluation is more than sufficient considering the space requirements.
- I liked the scheme used for the qualitative evaluation in Figure 1, displaying false positive and false negative pixels.


**Weaknesses:**

- The difference between the Dice to the manual and the semi-automatic segmentations is significant. This suggests that the semi-automatic segmentations are very different from the (manual) ground truth annotations. The proposed architecture, which is trained only on imperfect semi-supervised data, achieves a Dice of 62.07 to the manual annotations. It is unclear whether this performance is enough for the method to be usable in practice. Expanding the ablation study to make a comparison to the same method trained on wholly manual ground truth annotations would have been interesting.
- An additional comparison to another deep-learning self-supervised method would be nice. However, the current evaluation is extensive enough for a short paper.


**Deanonymize Review:**

no

**Detailed Comments:**

- The third sentence in the abstract is a bit too long and therefore difficult to understand. I would suggest splitting it into two sentences.
- It would be helpful for the reader to include an equation of the deformation-aware loss, if space permits.
- I would suggest rephrasing this sentence: “In addition to the final segmentation output, output of two of the following layers (different scales) interpolated using nearest-neighbour to the size of the segmentation mask are also considered for loss calculation - hence, multi-scale supervision.”
- It is unclear why the first table includes a standard deviation and the second table does not. It was also not clear to me whether the standard deviation refers to the deviation between subjects or between cross-validation folds. I would suggest merging both tables and using the additional space to expand the caption text.


**Justification Of The Rating:**

The authors make a methodological contribution by proposing the use of a deformation-aware self-supervision loss for the task of small vessel segmentation. The selected proxy task seems well-suited for the task at hand. The evaluation performed, including comparison to various baselines and a variant of the method, is sufficient for a short paper.

**Paper Type:**

methodological development

**Special Issue:**

no

---

### Official Review · Reviewer_8b3d · 2021-05-01

**Confidence:** 4
**Final Rating:** 3

**Summary:**

The paper proposes a deformation-aware semi-supervised learning model for small vessel segmentation with noisy training data. The paper uses the 7T MRI scanner to obtain the image, which is interesting as most of the segmentation tasks are conducted in images acquired in a 3T scanner.
The experimental results verify the effectiveness of the proposed method.

**Strengths:**

1. The proposed method utilizes the MRI images from a 7T scanner, which physically enables more detailed information on small vessels.
2. The proposed method achieves encouraging results compared to traditional methods.
3. According to the ablation study, the idea of combining multi-scale supervision and elastic deformation-based self-supervised learning does improve the segmentation performance.

**Weaknesses:**

1. Both multi-scale supervision and elastic deformation-based self-supervised learning are existing methods, thus the combination of the two methods are not sufficiently novel.
2. Though the authors claim the 7T MRI image can see better vessel details, the dice score seems far away from putting the method for real clinical practice.

**Deanonymize Review:**

no

**Detailed Comments:**

See strengths and weaknesses.

**Justification Of The Rating:**

The paper is well written, well structured and easy to follow. The paper includes an ablation study on the proposed methods. Though the novelty of combining existing methods is not sufficient, the results are encouraging.

**Paper Type:**

validation/application paper

**Special Issue:**

no

---

### Meta-Review · Area_Chair_Kw2f · 2021-05-09

**Recommendation:** Accept (Poster)
**Confidence:** 5

**Metareview:**

Strong contribution with promise for clinical impact. Some minor points should be improved (as outlined by reviews) otherwise the paper can be accepted for MIDL

---

### Decision · Program_Chairs · 2021-05-11

Accept (Poster)